# Smallpox outbreak scenarios and reactive intervention protocols: A mathematical model-based analysis applied to the Republic of Korea

Youngsuk Ko[1], Yubin Seo[2], Jin Ju Park[2], Eun Jung Kim[3], Jong Youn Moon[4], Tark Kim[5], Joong Sik Eom[6], Hong Sang Oh[7], Arim Kim[8], Jin Yong Kim[9], Jacob Lee[2]*, Eunok Jung[10]*

**1** Institute of Mathematical Sciences, Konkuk University, Seoul, Korea, **2** Division of Infectious Disease, Department of Internal Medicine, Kangnam Sacred Heart Hospital, College of Medicine, Hallym University, Seoul, Korea, **3** National Assembly Research Service, Seoul, Korea, **4** Department of preventive medicine, College of medicine, Gachon University, Incheon, Korea, **5** Division of Infectious Diseases, Soonchunhyang University Bucheon Hospital, Bucheon, Korea, **6** Division of Infectious Diseases, Department of Internal Medicine, Gil Medical Center, College of Medicine, Gachon University, Incheon, Korea, **7** Division of Infectious Disease, Department of Internal Medicine, Hallym University Sacred Heart Hospital, Gyeonggi-do, Korea, **8** Gachon Biomedical Convergence Institute, Korea, **9** Division of Infectious Diseases, Department of Internal Medicine, Incheon Medical Center, Incheon, Korea, **10** Department of Mathematics, Konkuk University, Seoul, Korea

* litjacob@gmail.com (JL); junge@konkuk.ac.kr (EJ)

## Abstract

Smallpox, caused by the variola virus, remains a potential biosecurity threat despite its eradication. This study develops a mathematical model to evaluate outbreak scenarios and the effectiveness of reactive intervention strategies in controlling transmission, with application to the Republic of Korea. The model incorporates age-stratified contact patterns, contact tracing, and vaccination strategies, including targeted vaccination and mass vaccination. Our simulations demonstrate that early outbreak recognition and rapid intervention are critical in mitigating smallpox spread. In scenarios where vaccination rollout was slow or outbreak recognition was delayed, severe patient numbers exceeded healthcare capacity, highlighting the need for pre-emptive preparedness. Sensitivity analyses revealed that outbreak recognition timing and contact tracing effectiveness were the most influential factors in determining outbreak severity, with later recognition leading to up to 3.5 times more cumulative cases. Furthermore, we compared different vaccination prioritization strategies and found that prioritizing high-transmission age groups was more effective in reducing total mortality than prioritizing high-risk groups based solely on disease severity. This contrasts with COVID-19 vaccination strategies, which focused on protecting vulnerable populations. These findings underscore the importance of early detection, strategic vaccination, and non-pharmaceutical interventions in mitigating a potential smallpox outbreak. Our model provides a quantitative framework for policymakers to evaluate intervention effectiveness and optimize outbreak response strategies.

**Data availability statement:** All relevant data supporting the findings of this study are publicly available in Figshare (https://doi.org/10.6084/m9.figshare.28329263.v5).

**Funding:** This research was supported by the Government-wide R&D Fund Project for Infectious Disease Research (GFID), Republic of Korea (grant No. HG23C1629). This work was supported by the Research Program funded by the Korea Disease Control and Prevention Agency (정책, 150). The funders had no role in study design, data collection and analysis, decision to publish, or preparation of the manuscript.".

**Competing interests:** The authors have declared that no competing interests exist.

**Abbreviations:** NPIs, Non-pharmaceutical interventions; PEP, Post-exposure prophylaxis; ROK, Republic of Korea; PI, Prediction interval; PRCC, Partial Rank Correlation Coefficient.

## Introduction

Smallpox, caused by the variola virus, is one of the most devastating diseases affecting humans [1]. With its origins traced back to ancient civilizations, smallpox has spread across continents. The disease is characterized by a high fever, severe skin eruptions, and a significant fatality rate (approximately 30%), making it a formidable threat to populations worldwide.

The pioneering findings of Jenner and concerted global efforts, including mass vaccination campaigns, surveillance, and implementation of ring vaccination strategies, led to the gradual decline of smallpox cases. Non-pharmaceutical interventions (NPIs), such as disease surveillance, case finding, and contact tracing, are also crucial for containing the spread of the virus [2]. Ring vaccination involves vaccinating all individuals near a detected case to prevent the spread of the virus. Additionally, post-exposure prophylaxis (PEP) has been used to vaccinate individuals exposed to the virus, further preventing outbreaks [3,4]. The World Health Organization (WHO) launched an intensified eradication program in 1967, which culminated in the declaration of smallpox eradication in 1980.

Despite its eradication, the variola virus remains in two high-security laboratories: the Centers for Disease Control and Prevention in the United States and the State Research Center of Virology and Biotechnology in Russia [5]. The virus is retained for research purposes, including developing new vaccines and treatments. However, the presence of these viral stocks poses biosecurity concerns, including the potential risk of bioterrorism. Given the lack of widespread immunity in the current global population, the accidental or deliberate release of the smallpox virus could lead to a catastrophic outbreak. Two notable laboratory-related smallpox incidents underscored the importance of maintaining vigilance. In 1971, the Aral Smallpox incident in Soviet Russia led to ten infections and three deaths [6]. Similarly, in 1978, a laboratory accident in Birmingham, England resulted in two infections and one death [7]. In recent years, additional laboratory-related incidents have raised concerns about smallpox security. In 2014, live smallpox vials were accidentally discovered at the U.S. Centers for Disease Control and Prevention (CDC) [8], and in 2021, unapproved vials labeled "VARIOLA" were unexpectedly found in a vaccine research facility in Pennsylvania [9]. Furthermore, advancements in synthetic biology have led to concerns about the feasibility of reconstructing the variola virus. In 2017, scientists successfully synthesized the horsepox virus, a close relative of smallpox, highlighting the potential for recreating smallpox through genetic engineering [10]. These incidents illustrate that despite eradication, the risk of smallpox re-emergence remains a global security and public health concern.

A notable scenario that underscores the potential threat of smallpox as a bioterrorism agent is Dark Winter exercises [11]. This senior-level bioterrorist attack simulation depicted a covert smallpox attack in the United States starting in Oklahoma City and rapidly spreading to other states. The exercise revealed significant gaps in the national emergency response, highlighting the challenges of containing the outbreak, managing public panic, and maintaining essential services. Winter's findings emphasize the

need for robust preparedness plans, including sufficient vaccine stockpiles, effective communication strategies, and coordinated efforts between public health and security agencies, to mitigate the impact of such a bioterrorism event. In recognition of the importance of healthcare system readiness, the Republic of Korea (ROK) has planned to expand its capacity for negative-pressure isolation beds to 3,500 to respond to Disease-X [12]. This expansion aims to strengthen the country's ability to respond to emerging infectious disease threats, ensuring that adequate hospital resources are available for managing highly transmissible and severe infections such as smallpox in the event of a bioterrorism attack.

Mathematical models are crucial in understanding and controlling infectious disease outbreaks, including smallpox, as demonstrated in various studies. Ferguson emphasized the effectiveness of targeted surveillance and containment interventions such as ring vaccination in controlling smallpox outbreaks, underscoring the need for a rapid response [13]. Meltzer constructed a model to evaluate quarantine and vaccination interventions after a bioterrorist attack and demonstrated that a combination of these strategies was effective in halting disease transmission [14]. Ohushima developed a model to predict smallpox outbreaks in Japan, evaluated the control measures, and found that mass vaccination was more effective than ring vaccination under certain conditions [15]. Chun used epidemic modeling and tabletop exercises to prepare public health officials in the ROK for potential outbreaks, highlighting the importance of these tools in estimating cases, deaths, and resource shortages [16]. Recent studies have expanded the scope of smallpox modeling, addressing various contexts and control strategies. Mohanty evaluated the impact of a hypothetical smallpox attack in India and assessed the effectiveness of different control measures, highlighting the importance of rapid response strategies in densely populated regions [17]. Similarly, Costantino investigated optimal vaccination strategies during a smallpox outbreak linked to bioterrorism, emphasizing the role of targeted vaccination campaigns in mitigating outbreak severity [18]. Several mathematical modeling studies have demonstrated the effectiveness of these interventions in mitigating disease spread. For instance, Rai analyzed the impact of social media advertisements on COVID-19 transmission dynamics, showing that public awareness campaigns significantly influenced behavioral changes and reduced infection rates [19]. Similarly, Sarkar developed a mathematical model to forecast COVID-19 transmission and demonstrated how social distancing and contact tracing played a crucial role in reducing case numbers across different regions [20].

In this study, we developed a mathematical model to analyze potential smallpox epidemics by incorporating various realistic factors such as age groups and heterogeneous contact patterns. The model includes contact tracing, targeted vaccination (which would mimic the behavior of ring vaccination), and mass vaccination strategies and distinguishes severe cases among infected individuals to discuss the required capacity for severe patients in emergency scenarios. The insights gained from vaccination prioritization during the COVID-19 pandemic were also analyzed in relation to potential smallpox scenarios. Given the potential biosecurity threat posed by smallpox, this study evaluated reactive intervention protocols by simulating various outbreak scenarios and assessing the effectiveness of NPIs, vaccination strategies, and isolation facilities. To enhance clarity, we define key epidemiological terms used in this study: "cases" refer to all infected individuals regardless of symptom severity, "confirmed cases" are those identified through clinical diagnosis or laboratory confirmation, "isolated patients" are confirmed cases placed under strict containment, "targeted vaccination" targets close contacts of confirmed cases, and "mass vaccination" refers to large-scale immunization of the general population regardless of exposure history. These findings highlight the importance of early detection, rapid response, and strategic vaccination prioritization. The proposed framework addresses smallpox and offers insights applicable to other emerging infectious diseases, emphasizing the need for comprehensive preparedness in the face of potential outbreaks.

## Materials and methods

### Control measures

This study incorporates the following intervention strategies to analyze their impact on smallpox outbreak dynamics. Details on how these intervention strategies are incorporated into the model are provided in the following sections.

- Contact tracing: Identification and isolation of individuals who have had close contact with confirmed cases to prevent further transmission.

- Social distancing: Reduction of overall contact number in the population to limit the spread of infection.

- Targeted vaccination: In response to confirmed cases, a proportion of susceptible individuals in the population is vaccinated to simulate the effect of a containment-focused vaccination strategy. In our ODE-based model, this is implemented by prioritizing a subset of the population for immediate vaccination following outbreak recognition.

- Mass vaccination: A broad immunization campaign administered across the population, independent of exposure history. This strategy is modeled as a gradual increase in vaccination coverage following a logistic growth function, accounting for vaccine distribution constraints.

## Mathematical modeling of smallpox epidemic

To investigate the transmission dynamics of potential smallpox epidemics, we developed a susceptible-infectious-recovered-type mathematical model that reflects contact tracing, disease severity, age group, and targeted/mass vaccination. The mathematical model is shown as a flow diagram in Fig 1. Age groups are denoted by subscript ($i$) for each variable, and vaccinated individuals are denoted by superscript ($v$). In this study, we categorized the population into 16 age groups, each spanning 5 years, ranging from 0–4 years to 75 years and older in the Republic of Korea [21].

In general, Susceptible-Infectious-Recovered type models have the transmission rate, typically denoted by the symbol $\beta$, which consists of the contact rate per unit time ($c$) and the probability of successful disease transmission ($p$), that is, $\beta = pc$. In this study, we distinguished between those who had contact with infectious hosts but were not infected ($C_i$) and those who were infected after contact ($E_i$). We also considered both close contact ($\lambda_i^A$) and social contact ($\lambda_i^B$). Close

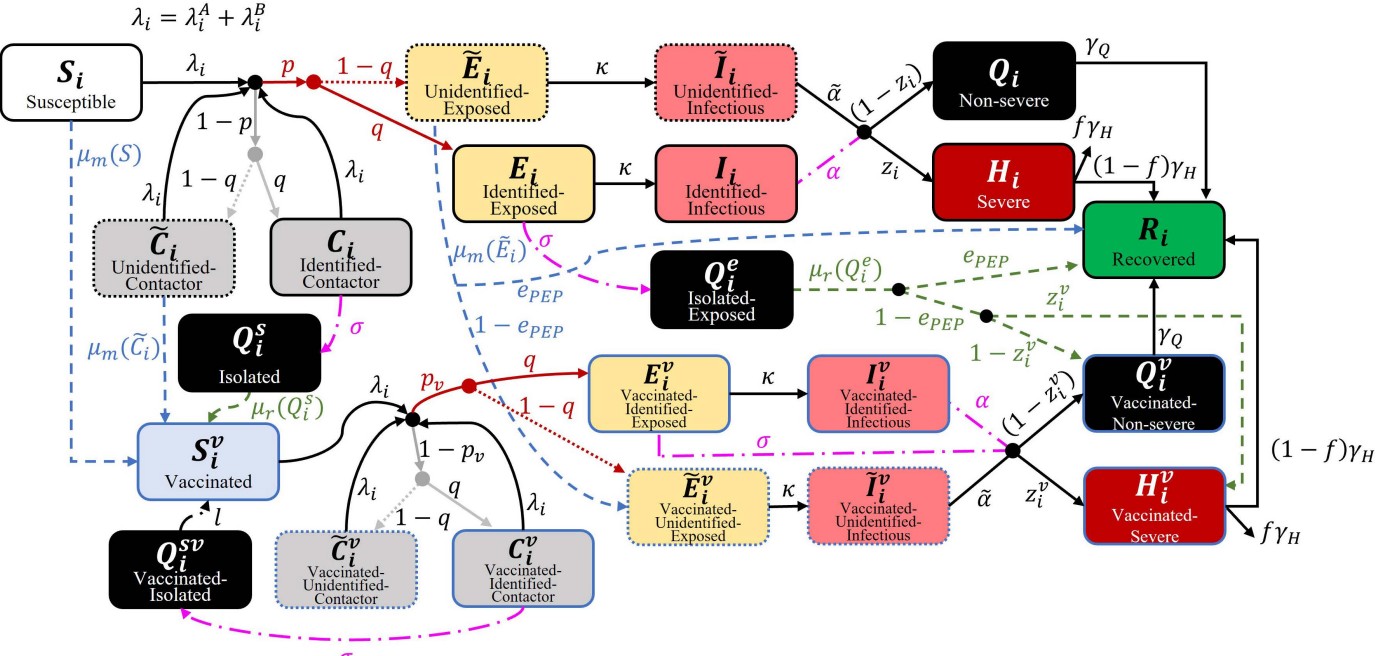

**Fig 1. Flow diagram of the mathematical model of the smallpox epidemic.**

contacts were incorporated based on a study by Prem et al.[22]. Social contacts were assumed to be four times the number of contacts, excluding household contacts. The probability of successful disease transmission through close contact was set to 60% [23]. Fig 2 visualizes close contacts and social contacts using heatmap graphs. Using this value and the next-generation matrix method, the basic reproductive number through close contact was calculated to be approximately four [24]. Considering that the recorded basic reproductive number for smallpox is around six, this suggests that social contacts contribute about two. To reflect this, we assumed that social contacts occur at four times the frequency of close contacts and set the probability of transmission through social contact at 10%, based on this frequency difference [25]. These parameter values were chosen as simplifying assumptions to reproduce an overall basic reproductive number of approximately six, rather than as precise empirical estimates of contact-specific transmission probabilities.

To incorporate contact tracing regardless of infection status, we used the symbol ($q$) to represent the proportion of contact-traced individuals. Those who were traced after contact and those who were not traced were distinguished using a tilde symbol ($\sim$). Here, casual contact was not traced. The infection transmission periods for traced and non-traced individuals were set based on the time from symptom onset to isolation during the COVID-19 pandemic in the ROK [26]. We assumed that contact-traced individuals would be hospitalized/isolated relatively quickly (within 2.3 days, $1/\alpha$) after symptom onset, while non-traced individuals would be isolated after 6.8 days ($1/\widetilde{\alpha}$), considering the time to the appearance of definitive smallpox symptoms (lesions) [27]. The severity rate in the model ($z_i$) was set to twice the infection fatality ratio. Thus, in the model simulations, half of the severe patients died, whereas there were no deaths among non-severe patients.

Vaccination was applied in two forms in the model: mass vaccination ($\mu_m$) and targeted vaccination ($\mu_r$). Targeted vaccination was modeled and applied following outbreak recognition. The number of individuals requiring immediate vaccination was determined based on the distribution of contagious contacts at the outbreak recognition time point. Targeted vaccination was prioritized and administered after case isolation but not in those who had already developed symptoms. Patients exposed to the infection either recovered or continued to show symptoms depending on the effectiveness of PEP ($e_{PEP}$). Even those who did not directly receive the effects of PEP experienced a reduction in severity/fatality rates due to partial effects. Mass vaccination was administered to the entire population outside the targeted vaccination, and those who had already been vaccinated were not revaccinated even if they were contact-traced. Those who were isolated without infection or targeted vaccination were discharged 19 days ($1/l$) after isolation [28].

In response to the smallpox epidemic, we assumed that vaccinating 40 million people (approximately 80% of the population) would be necessary to achieve herd immunity. Vaccination can be slow initially owing to the lack of available

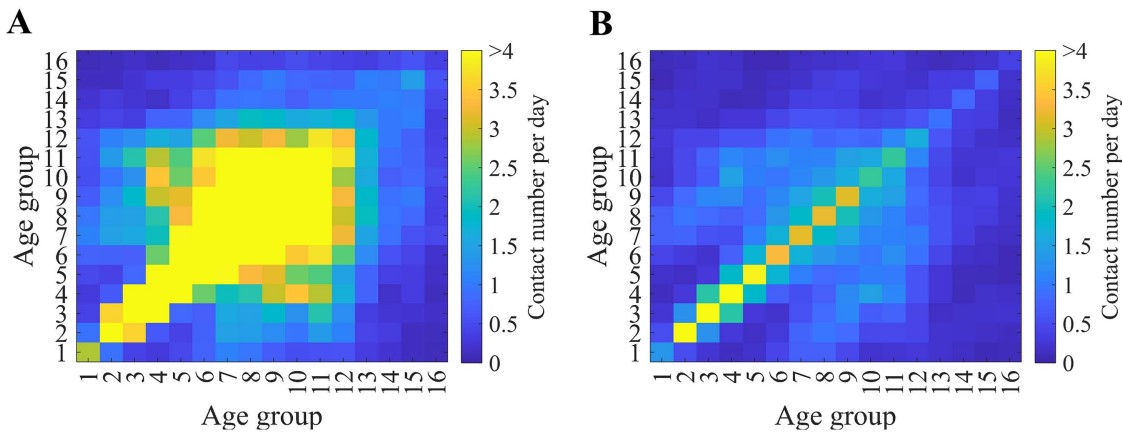

**Fig 2. Contact matrices used for the model simulation: Close contact (A) and social contact (B).**

personnel who are educated and vaccinated. In the ROK, there was a small-scale vaccination for healthcare workers during the global Mpox outbreak in 2022 in response to the domestic influx [29]. These individuals would be the first to start vaccination during a smallpox epidemic and could vaccinate other healthcare workers and target populations. Thus, we assumed the daily vaccination capacity follows a logistic growth model, considering the maximum daily vaccination capacity during the COVID-19 pandemic, set at 1 million ($\mu_{ub}$). The vaccination process started with 1,000 vaccinations per day, with a logistic growth rate ($r_v$) of 0.1, and all 40 million people were vaccinated after 110 days.

The model is formulated using ordinary differential equations as follows:

$$\frac{dS_i}{dt} = -\left(\lambda_i^A + \lambda_i^B\right) S_i - \mu_m(S) , \quad \frac{dS_i^v}{dt} = \mu_m(S) + \mu_m\left(\widetilde{C}_i\right) + \mu_r\left(Q_i^s\right) + IQ_i^{sv} - \left(\lambda_i^A + \lambda_i^B\right) S_i^v,$$

$$\frac{d\widetilde{C}_i}{dt} = \left(1 - p^A\right)\left(1 - q\right)\lambda_i^A\left(S_i + \widetilde{C}_i\right) + \left(1 - p^B\right)\lambda_i^B\left(S_i + \widetilde{C}_i\right) - \left(\lambda_i^A + \lambda_i^B\right)\widetilde{C}_i - \mu_m\left(\widetilde{C}_i\right),$$

$$\frac{dC_i}{dt} = \left(1 - p^A\right) q\lambda_i^A\left(S_i + \widetilde{C}_i\right) + \left(1 - p^A\right)\lambda_i^A C_i + \left(1 - p^B\right)\lambda_i^B C_i - \left(\lambda_i^A + \lambda_i^B\right) C_i - \sigma C_i ,$$

$$\frac{d\widetilde{C}_i^v}{dt} = \left(1 - p_v^A\right)\left(1 - q\right)\lambda_i^A\left(S_i^v + \widetilde{C}_i^v\right) + \left(1 - p_v^B\right)\lambda_i^B\left(S_i^v + \widetilde{C}_i^v\right) - \left(\lambda_i^A + \lambda_i^B\right)\widetilde{C}_i^v,$$

$$\frac{dC_i^v}{dt} = \left(1 - p_v^A\right) q\lambda_i^A\left(S_i^v + \widetilde{C}_i^v\right) + \left(1 - p_v^A\right)\lambda_i^A C_i^v + \left(1 - p_v^B\right)\lambda_i^B C_i^v - \left(\lambda_i^A + \lambda_i^B\right) C_i^v - \sigma C_i^v,$$

$$\frac{d\widetilde{E}_i}{dt} = p^A(1 - q)\lambda_i^A\left(S_i + \widetilde{C}_i\right) + p^B\lambda_i^B\left(S_i + \widetilde{C}_i\right) - \kappa\widetilde{E}_i - \mu_m\left(\widetilde{E}_i\right),$$

$$\frac{dE_i}{dt} = p^A q\lambda_i^A\left(S_i + \widetilde{C}_i\right) + p^A\lambda_i^A C_i + p^B\lambda_i^B C_i - \kappa E_i - \sigma E_i ,$$

$$\frac{d\widetilde{E}_i^v}{dt} = (1 - e_{pep})\mu_m\left(\widetilde{E}_i\right) + p_v^A(1 - q)\lambda_i^A\left(S_i^v + \widetilde{C}_i^v\right) + p_v^B\lambda_i^B\left(S_i^v + \widetilde{C}_i^v\right) - \kappa\widetilde{E}_i^v,$$

$$\frac{dE_i^v}{dt} = p_v^A q\lambda_i^A\left(S_i^v + \widetilde{C}_i^v\right) + p_v^A\lambda_i^A C_i^v + p_v^B\lambda_i^B C_i^v - \kappa E_i^v - \sigma E_i^v ,$$

$$\frac{d\widetilde{I}_i}{dt} = \kappa\widetilde{E}_i - \widetilde{\alpha}\widetilde{I}_i , \quad \frac{dI_i}{dt} = \kappa E_i - \alpha I_i ,$$

$$\frac{d\widetilde{I}_i^v}{dt} = \kappa\widetilde{E}_i^v - \widetilde{\alpha}\widetilde{I}_i^v , \quad \frac{dI_i^v}{dt} = \kappa E_i^v - \alpha I_i^v ,$$

$$\frac{dQ_i^s}{dt} = \sigma C_i - \mu_r\left(Q_i^s\right) , \quad \frac{dQ_i^{sv}}{dt} = \sigma C_i^v - IQ_i^{sv}, \quad \frac{dQ_i^e}{dt} = \sigma E_i - \mu_r\left(Q_i^e\right) ,$$

$$\frac{dQ_i}{dt} = (1 - z_i)\left(\widetilde{\alpha}\widetilde{I}_i + \alpha I_i\right) - \gamma_Q Q_i,$$

$$\frac{dQ_i^v}{dt} = (1 - z_i^v)\left((1 - e_{PEP})\mu_r\left(Q_i^e\right) + \widetilde{\alpha}\widetilde{I}_i^v + \alpha I_i^v + \sigma E_i^v\right) - \gamma_Q Q_i^v,$$

$$\frac{dH_i}{dt} = z_i \left( \widetilde{\alpha} \widetilde{I}_i + \alpha I_i \right) - \gamma_H H_i,$$

$$\frac{dH_i^v}{dt} = z_i^v \left( (1 - e_{PEP}) \mu_r (Q_i^e) + \widetilde{\alpha} \widetilde{I}_i^v + \alpha I_i^v + \sigma E_i^v \right) - \gamma_H H_i^v,$$

$$\frac{dR}{dt} = e_{PEP} \left( \mu_m \left( \widetilde{E}_i \right) + \mu_r (Q_i^e) \right) + \gamma_Q (Q_i + Q_i^v) + (1 - f) \gamma_H (H_i + H_i^v),$$

$$\lambda_i^A = \sum_j \frac{c_{ij}^A \left( I_j + I_j^v + \widetilde{I}_j + \widetilde{I}_j^v \right)}{N}, \quad \lambda_i^B = \sum_j \frac{c_{ij}^B \left( I_j + I_j^v + \widetilde{I}_j + \widetilde{I}_j^v \right)}{N},$$

$$N = \sum_i S_i + S_i^v + \widetilde{C}_i + C_i + \widetilde{C}_i^v + C_i^v + \widetilde{E}_i + E_i + \widetilde{E}_i^v + E_i^v + \widetilde{I}_i + I_i + \widetilde{I}_i^v + I_i^v + R_i,$$

$$\mu(t) = \frac{\mu_{ub}}{1 + \exp \left( -r_v (t - t_0) \right)},$$

$$\mu_r^* = \min \left( \mu(t), \sum_i Q_i^s + Q_i^e \right),$$

$$\mu_r(X) = \mu_r^* \frac{X}{\sum_i Q_i^s + Q_i^e},$$

$$\mu_m(X) = (\mu(t) - \mu_r^*) \frac{X}{\sum_i S_i + \widetilde{C}_i + \widetilde{E}_i}.$$

The model parameters are listed in Table 1. As experienced during the COVID-19 pandemic, the scale of the potential outbreaks remains uncertain. To reflect this, model simulations were conducted as a stochastic process using the Tau-leaping method. We ran 1000 simulations for each model setting using MATLAB 2024b. As an initial condition for the model simulation, all population groups were assumed to be in a susceptible state. The number of exposed hosts was fixed at 100 and was proportionally distributed among age groups based on their population sizes, ensuring that the total remained constant at 100. This distribution was deterministic and remained the same across all simulations, rather than being randomly assigned. Fig 3 visualizes the population numbers in each age group.

## Baseline model simulation scenario

The simulation time was set as 365 days. The baseline model simulation scenario consisted of three phases following the initial exposure:

- Pre-declaration (Phase 1): This phase represents the period during which the occurrence of the outbreak has not yet been recognized. No NPIs or vaccination measures were in place during this phase, which lasted for 28 days after the initial exposure. This is the worst-case scenario, set at a realistic level, considering the incubation period, initial symptoms, and occurrence of rashes.

- Post-declaration (Phase 2): This phase begins when the outbreak is first recognized, and contact tracing and social distancing (NPIs) are implemented. However, vaccination is not yet feasible because of the time required for preparation,

**Table 1. Model parameters.**

| Symbol | Description | Value | Reference |
|---|---|---|---|
| $1/\tilde{\alpha}$ | Infectious period of contact-unidentified hosts | 6.8 days | [26,27] |
| $1/\alpha$ | Infectious period of contact-identified hosts | 2.3 days | [26] |
| $p_A$ | Probability of successful disease transmission per close contact | 0.6 | [23, 24, 26] |
| $p_B$ | Probability of successful disease transmission per casual contact | 0.1 | [23, 24, 26] |
| $q$ | Contact-identification ratio | 0.8 | Assumed |
| $1/l$ | Isolation duration of uninfected case | 19 days | [28] |
| $1/\sigma$ | Contact tracing duration | 2 days | Assumed |
| $1/\kappa$ | Incubation period | 12 days | [30] |
| $1/\gamma_Q$ | Isolation duration of non-severe case | 28 days | [30] |
| $1/\gamma_H$ | Duration from hospitalization to recovery (or death) of severe case | 13 days | [1] |
| $z_i$ | Age dependent severity rate | 0.83 (1)<br>0.47 (2,3)<br>0.61 (4,5)<br>0.59 (6789–10)<br>0.64 (1112131415–16) | [31] |
| $f$ | Fatality rate of severe case | 0.5 | Assumed, [1] |
| $e$ | Preventive vaccine effectiveness against infection | 0.78 | [32] |
| $e_H$ | Vaccine effectiveness against severity | 0.97 | [32] |
| $e_{pep}$ | Post-exposure prophylaxis vaccine effectiveness against infection | 0.5 | [17] |

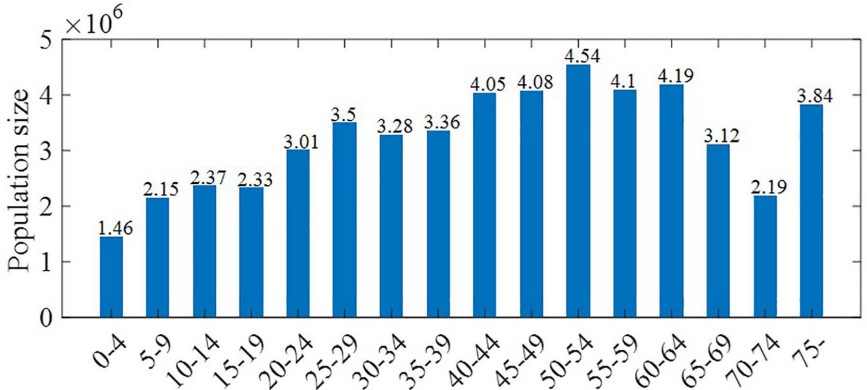

**Fig 3. Age-group population distribution used in the model.** The horizontal bar graph represents the population size for each age group. The numbers above the bars indicate the population size in millions (e.g., 3.5 represents 3.5 million people).

e.g., training for medical personnel. This phase lasted for three days. It was assumed that social distancing would reduce the total contact rate by 60%.

- Post-vaccination (Phase 3): This phase marks the beginning of vaccination. Vaccination continued until 40 million people had been vaccinated. This phase lasted for 334 days. Vaccination was administered simultaneously to all age groups, and the amount of vaccines administered was proportional to the population of each group; that is, there was no vaccine prioritization.

## Scenarios considering vaccine prioritization

Vaccination was not prioritized in the baseline scenario considered in this study. However, as observed during the COVID-19 pandemic, strategic prioritization of vaccine distribution can be crucial for outbreak control. To explore the impact of different prioritization strategies, we implemented four distinct vaccination prioritization criteria:

1. Ascending age – Vaccination is first administered to the youngest age group and progresses sequentially toward older age groups.

2. Descending age – Vaccination is first administered to the oldest age group and progresses sequentially toward younger age groups.

3. Prioritizing age groups with higher transmission risk

4. Prioritizing age groups with higher severity/death risk – Vaccination is allocated first to age groups with the highest probability of severe disease and mortality.

The age groups with higher transmission risk were determined by calculating the column sum of the next-generation matrix, which represents the total potential transmission risk each age group poses to others. Since the probability of successful disease transmission per close or social contact does not vary by age group in our model, the ranking is proportional to the contact rates and population size, and the order was as follows: age groups 4 (15–19 year), 3 (10–14 year), 9 (40–44 year), 8 (35–39 year), 7 (30–34 year), 6 (25–29 year), 10 (45–49 year), 11 (50–54 year), 5 (20–24 year), 2 (5–9 year), 12 (55–59 year), 13 (60–64 year), 1 (0–4 year), 14 (65–69 year), 15 (70–74 year), 16 (75 + year).

## Sensitivity analysis

To address the inherent uncertainties in these values and conduct a comprehensive sensitivity analysis of the model outcomes, we measured the partial rank correlation coefficient (PRCC) values using Latin hypercube sampling. PRCC is a statistical measure used to determine the strength and direction of the relationship between two variables while controlling for the effects of other variables. In sensitivity analysis, it is particularly useful to identify the parameters that have the most significant impact on the output of a model. A detailed description of this method is provided in reference [33]. We considered the following model inputs:

- Outbreak recognition timing – This parameter determines when interventions begin by setting the delay between the initial exposure and the implementation of control measures. A shorter recognition time allows for earlier contact tracing, isolation, and vaccination.

- Impact of social distancing on contact number – This represents the effectiveness of social distancing measures in reducing population-wide contact rates. Higher values indicate greater reductions in interpersonal interactions, which directly affect the force of infection in the model by lowering transmission opportunities.

- Infectious period of traced and non-traced cases – These parameters define the duration of infectiousness before an individual is isolated. Traced cases (identified through contact tracing) have a shorter infectious period due to faster isolation, whereas non-traced cases remain infectious for a longer duration, increasing the likelihood of further transmission.

- Contact identification ratio – This parameter determines the proportion of an infected individual's contacts that are successfully traced and quarantined. Higher values enhance the effectiveness of contact tracing, reducing the number of secondary infections.

- Logistic growth rate of the daily vaccination number – This parameter governs the rate at which vaccination capacity scales up over time. A higher growth rate enables faster vaccine distribution, accelerating immunity buildup and controlling the outbreak more efficiently.

Latin hypercube sampling was used to generate samples from parameter ranges informed by ±25% variability around baseline values. PRCC values were calculated over time to evaluate the relative importance of each parameter in influencing the model outputs. The model outputs were set as the cumulative confirmed cases and deaths, and the peak number of administered severe patients.

## Results

### Baseline scenario simulation: Outbreak outcomes

Fig 4 shows the model simulation results for the confirmed cases ($\sum_i \widetilde{\alpha}\widetilde{I}_i + \alpha I_i$, transition from infectious to isolated) and isolated patients ($\sum_i \left[ \widetilde{\alpha} \left( \widetilde{I}_j + \widetilde{I}_j^v \right) + \alpha \left( I_j + I_j^v \right) \right]$). Daily confirmed cases increased sharply (mean 1167, maximum 1602 in 95% prediction interval; PI) owing to contact tracing implemented at the initial outbreak recognition, and then decreased, followed by a gradual increase, and finally decreased again owing to herd immunity from vaccination. The number of non-severe patients ($\sum_i \left( Q_i + Q_i^v \right)$) reached a mean of 3750 (maximum 5422 in 95% PI) after 114 days of spread, whereas that of severe patients ($\sum_i \left( H_i + H_i^v \right)$) reached a mean of 1235 (maximum 1838 in 95% PI) after 99 days of exposure.

Fig 5 presents the distribution of confirmed cases and deaths from the model simulations across age groups using box-and-whisker plots. Panel A shows the total number of confirmed cases across all age groups, while Panel B displays the age-specific distribution of confirmed cases. Similarly, Panel C illustrates the total number of deaths, and Panel D presents the age-specific death distribution. The total number of confirmed cases was 36,600 (95% PI [24,253, 51,500]), and the total number of deaths was 5,345 (95% PI [3,472, 7,545]). Age group 9 (40–44 years) had the highest number of confirmed cases, with a mean of 4,241 (95% PI [2,846, 5,969]), while age group 11 (50–54 years) had the highest number of deaths, with a mean of 677 (95% PI [447, 936]). The 0–4 years age group had the lowest number of confirmed cases (mean 361, 95% PI [237, 513]) and deaths (mean 68, 95% PI [41, 101]).

In addition to the baseline scenario, Fig 6 visualizes the distribution of outbreak outcomes according to the vaccination growth rate and the reduction in contacts due to social distancing, assuming an initial vaccination number of 1,000. Tables 2–4 present the results for additional scenarios, including initial vaccination numbers of 1,000, 5,000, and 10,000. When contact reduction increased from 50% to 70%, total confirmed cases decreased substantially, from 90,947–17,565 under the baseline vaccination scenario (1,000 initial vaccinations, growth rate = 0.1). Similarly, total deaths declined from 11,890–2,851. Faster

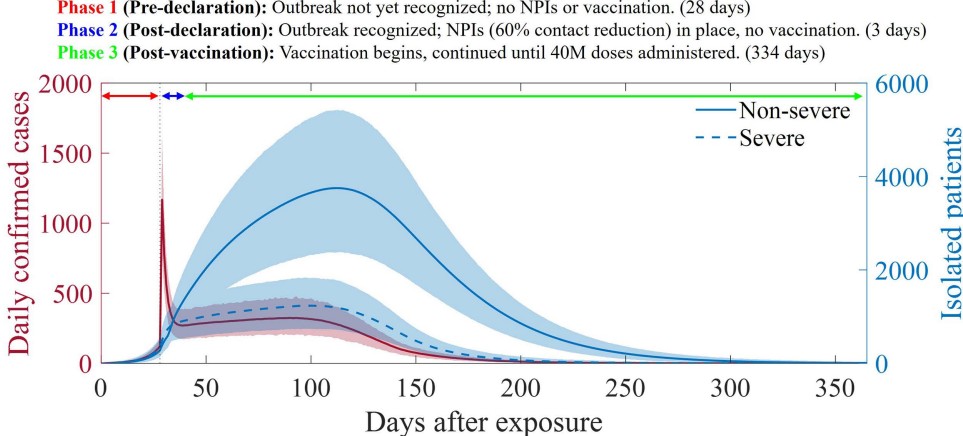

**Fig 4. Baseline scenario simulation results.** The curves represent the mean values of the model simulations, and shaded areas indicate the 95% prediction interval. The red graph shows the daily confirmed cases, whereas the blue curves represent isolated patients. Among the blue curves, the solid and dashed lines indicate patients with non-severe and severe patients.

**Fig 5. Outbreak outcomes from the baseline scenario simulations.** Panels (A) and (B) show the distribution of confirmed cases by age group and in total, respectively. Panels (C) and (D) present the corresponding distributions of deaths. Results are displayed as violin plots, which illustrate both the variability and the probability density of outcomes across stochastic simulation runs.

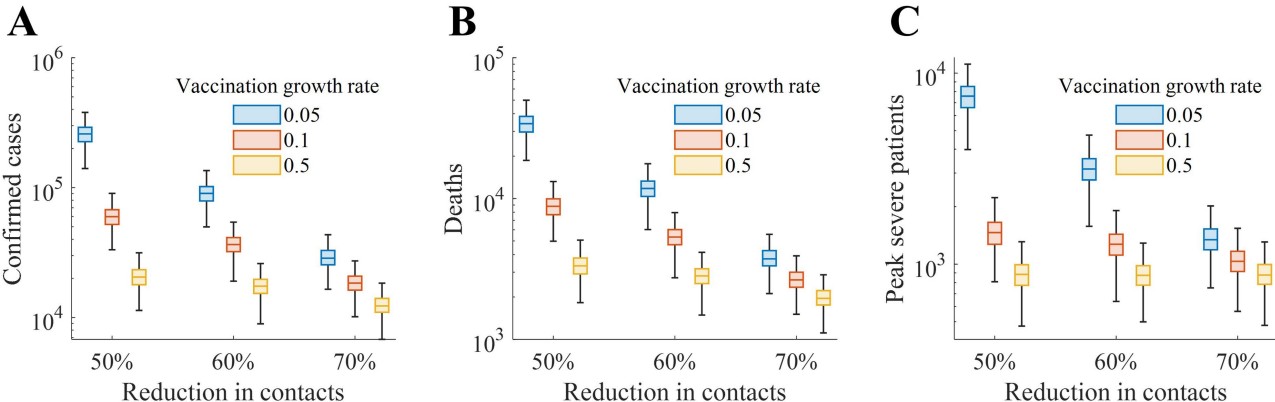

**Fig 6. Outbreak outcomes under different assumptions for vaccination growth rate and contact reduction, given a baseline initial vaccination number of 1,000.** Results are shown on a log scale for confirmed cases (A), deaths (B), and peak severe patients (C). Each color denotes a different vaccination growth rate, while the horizontal axis indicates the level of contact reduction.

**Table 2. Mean and 95% prediction interval of outbreak outcomes with a 50% reduction in contacts due to social distancing.**

| Initial vaccination number | Growth rate | Confirmed cases | Deaths | Peak severe patients |
|---|---|---|---|---|
| 1000 (baseline) | 0.05 | 260458 [170462, 354688] | 34255 [22388, 46734] | 7608 [4983, 10423] |
| | 0.1 (baseline) | 90947 [60018, 127048] | 11890 [7842, 16516] | 3182 [2080, 4452] |
| | 0.5 | 29323 [19519, 40916] | 3816 [2588, 5298] | 1369 [909, 1915] |
| 5000 | 0.05 | 144457 [94173, 199367] | 19008 [12540, 25960] | 4411 [2875, 6109] |
| | 0.1 (baseline) | 64585 [41862, 90666] | 8451 [5588, 11793] | 2381 [1562, 3349] |
| | 0.5 | 26828 [17542, 37368] | 3491 [2316, 4847] | 1282 [849, 1796] |
| 10000 | 0.05 | 111421 [73076, 157301] | 14671 [9640, 20653] | 3507 [2303, 4973] |
| | 0.1 (baseline) | 56267 [38326, 76475] | 7358 [5030, 9959] | 2135 [1458, 2922] |
| | 0.5 | 26075 [17039, 37098] | 3391 [2269, 4682] | 1261 [819, 1759] |

**Table 3. Mean and 95% prediction interval of outbreak outcomes with a 60% reduction in contacts due to social distancing.**

| Initial vaccination number | Growth rate | Confirmed cases | Deaths | Peak severe patients |
|---|---|---|---|---|
| 1000 (baseline) | 0.05 | 60356 [40271, 84492] | 8894 [5950, 12436] | 1475 [985, 2094] |
| | 0.1 (baseline) | 36886 [24398, 51467] | 5391 [3581, 7513] | 1281 [838, 1786] |
| | 0.5 | 18611 [12524, 25865] | 2676 [1827, 3706] | 1048 [707, 1455] |
| 5000 | 0.05 | 46071 [30001, 63246] | 6784 [4461, 9343] | 1318 [848, 1808] |
| | 0.1 (baseline) | 30803 [20538, 42278] | 4495 [2996, 6173] | 1208 [797, 1682] |
| | 0.5 | 17614 [11806, 24104] | 2530 [1713, 3410] | 1028 [691, 1411] |
| 10000 | 0.05 | 40645 [26835, 56652] | 5981 [3918, 8371] | 1261 [827, 1787] |
| | 0.1 (baseline) | 28100 [19317, 39537] | 4098 [2792, 5702] | 1168 [790, 1642] |
| | 0.5 | 17103 [11403, 23652] | 2457 [1669, 3365] | 1012 [684, 1404] |

**Table 4. Mean and 95% prediction interval of outbreak outcomes with a 70% reduction in contacts due to social distancing.**

| Initial vaccination number | Growth rate | Confirmed cases | Deaths | Peak severe patients |
|---|---|---|---|---|
| 1000 (baseline) | 0.05 | 20749 [13713, 29494] | 3370 [2225, 4805] | 894 [602, 1256] |
| | 0.1 (baseline) | 17565 [11933, 23971] | 2851 [1947, 3852] | 881 [605, 1186] |
| | 0.5 | 12510 [8588, 17000] | 1985 [1358, 2681] | 890 [611, 1224] |
| 5000 | 0.05 | 19069 [13225, 26800] | 3101 [2105, 4343] | 891 [625, 1237] |
| | 0.1 (baseline) | 16446 [11093, 22420] | 2663 [1853, 3608] | 893 [596, 1228] |
| | 0.5 | 12006 [8172, 16475] | 1898 [1321, 2584] | 884 [598, 1215] |
| 10000 | 0.05 | 18288 [12598, 25139] | 2975 [2042, 4084] | 891 [625, 1210] |
| | 0.1 (baseline) | 15724 [11007, 21809] | 2545 [1781, 3533] | 889 [609, 1224] |
| | 0.5 | 11900 [8028, 16371] | 1879 [1285, 2577] | 890 [596, 1228] |

vaccine deployment also had a significant impact; for instance, under 50% contact reduction, increasing the vaccination growth rate from 0.05 to 0.5 reduced confirmed cases from 260,458–29,323 and deaths from 34,255–3,816.

## Baseline scenario simulation: Impact of outbreak recognition on outbreak size

Fig 7 shows the distribution of hosts who had contact with infected individuals at outbreak recognition 28 days after initial exposure. The average numbers of hosts who had contact (whether they were infected or not), hosts during the incubation period, and hosts in the contagious stage were 5618 (95% PI [3786, 7776]), 3401 (95% PI [2288, 4706]), and 926 (95% PI [623, 1275]), respectively.

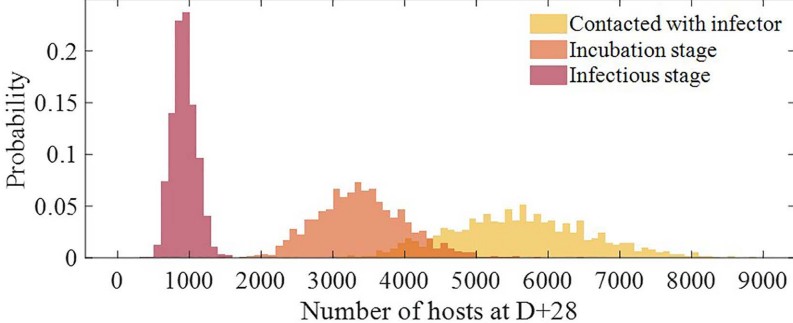

**Fig 7. Distribution of the number of hosts who had hazardous contact until the outbreak recognition in different stages.**

## Baseline scenario simulation: Vaccine administration over time

Fig 8 shows the mean daily vaccination number (Panel A), the distribution of administered targeted vaccinations in the simulation runs (Panel B), and the duration of the targeted vaccination period (Panel C). The mean daily vaccination number refers to the average number of vaccines administered per day across all simulations, which varies for targeted vaccination due to its dependency on the number of identified contacts in each run. In contrast, population-wide vaccination follows a fixed schedule and remains constant across simulations. In this study, the targeted vaccination period was considered the point at which the mass vaccination exceeded the amount of targeted vaccination. The average number of targeted vaccinations administered was 28750 (95% PI [19145,40218]), and the vaccination period lasted an average of 5.53 days (95% PI [3.5, 8]).

## Impact of vaccine prioritization

The simulation results, including the baseline scenario and scenarios with vaccine prioritization, are shown in Fig 9. Panels A and B show the daily numbers of confirmed and severe cases, respectively. Compared with the baseline scenario, prioritizing vaccination for age groups with a higher transmission risk (purple) showed a decrease, whereas prioritizing vaccination for older age groups (yellow) showed a significant increase. Table 5 lists the odds ratios for cumulative confirmed cases, deaths, and the peak number of severe patients compared to the baseline scenario. When prioritization based on transmissibility was applied, the odds ratios for all metrics were below one, within the confidence interval. Conversely, when prioritization based on descending age was applied, the odds ratios for all metrics exceeded one.

## Sensitivity analysis: Factors with changing influence over time

We conducted a sensitivity analysis to examine which parameters most influenced outbreak outcomes over time (see Materials and Methods section for details). Fig 10 shows the measured absolute values of PRCC over time. The order of the absolute PRCC values for the model inputs was the same regardless of whether the model output was cumulative confirmed cases or deaths. Based on the final absolute PRCC values, the timing of outbreak recognition had the highest value, with 0.90 for cumulative confirmed cases and deaths. The contact identification ratio initially had a relatively high absolute PRCC (0.29) for cumulative confirmed cases but decreased to 0.01, resulting in the lowest PRCC. Table 6 lists the ranges of PRCC values. In contrast to Fig 9, the absolute value of the PRCC for the growth rate of daily vaccination was the second smallest when the considered model output peaked for severe patients. Additionally, outbreak recognition timing had the highest absolute PRCC.

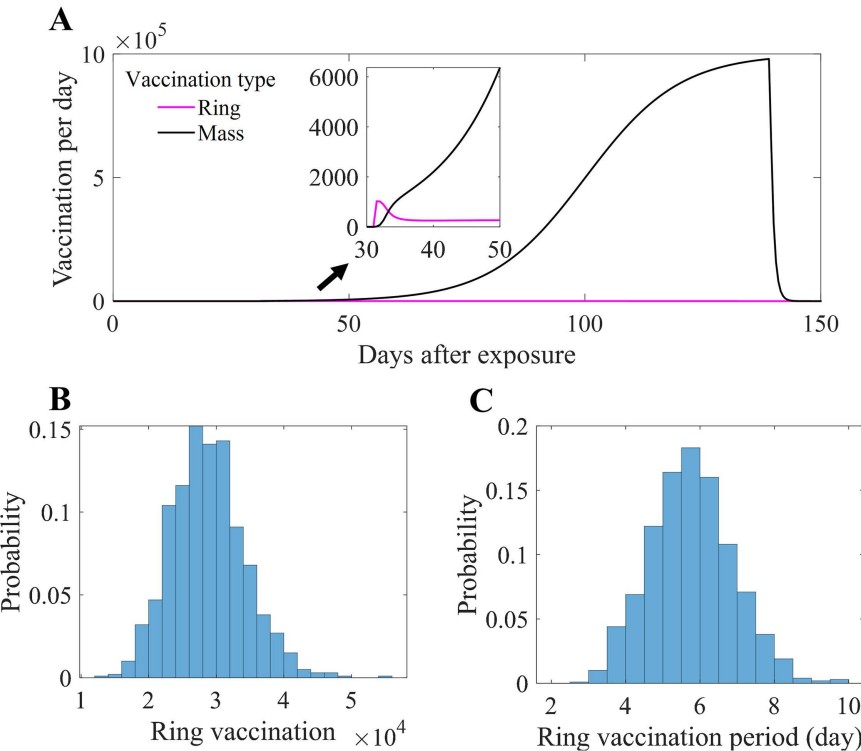

Fig 8. Vaccination numbers in the baseline scenario simulations. Panel (A) shows the mean daily number of vaccinations for ring vaccination (magenta) and mass vaccination (black), with the inset highlighting the early outbreak period. Panels (B) and (C) present the probability distributions for the total number of doses administered and the duration of the ring vaccination campaign, respectively, across stochastic simulation runs.

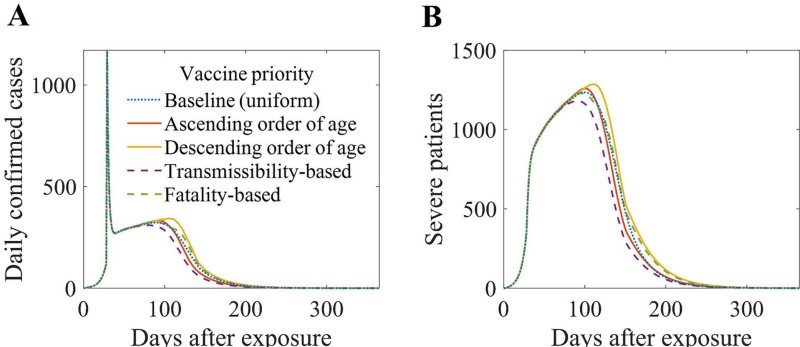

Fig 9. Simulation results under different vaccine prioritization strategies. Panel (A) shows daily confirmed cases, and Panel (B) shows the number of severe patients over time. The scenarios compared are: baseline (uniform vaccination, dotted blue), ascending order of age (solid orange), descending order of age (solid yellow), transmissibility-based prioritization (dashed purple), and fatality-based prioritization (dashed green).

## Discussion

Our findings align with previous modeling studies emphasizing early outbreak recognition and rapid intervention as key factors in controlling smallpox outbreaks. Ferguson et al. and Meltzer et al. demonstrated the effectiveness of targeted

**Table 5. Odds ratios and confidence intervals for scenarios considering vaccine prioritization compared to the baseline scenario.**

| Vaccine prioritization | Confirmed cases | Deaths | Peak number of administered patients |
|---|---|---|---|
| Ascending age | 0.97, [0.95 0.98] | 0.98, [0.96 0.99] | 1.02, [1.00 1.03] |
| Descending age | 1.07, [1.06 1.09] | 1.08, [1.06 1.09] | 1.04, [1.02 1.06] |
| Transmissibility | 0.89, [0.87 0.90] | 0.90, [0.89 0.92] | 0.96, [0.94 0.97] |
| Fatality | 1.03, [1.02 1.05] | 1.03, [1.01 1.05] | 1.00, [0.98 1.01] |

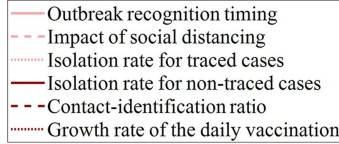

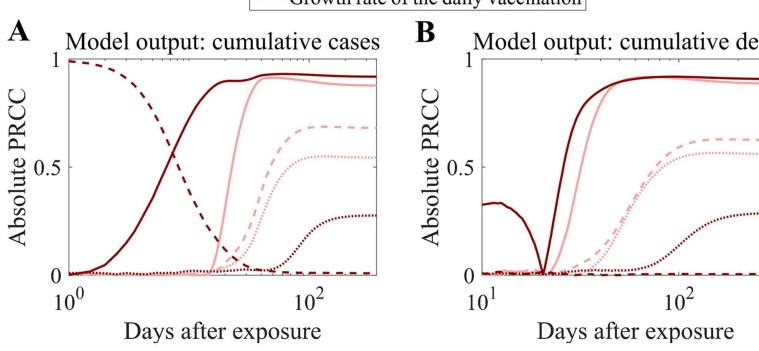

**Fig 10. Absolute partial rank correlation coefficients (PRCC) over time for key model parameters.** Panel (A) shows PRCC values with respect to cumulative confirmed cases, and Panel (B) shows PRCC values with respect to cumulative deaths. Each color denotes a different input parameter: outbreak recognition timing (yellow), impact of social distancing (green), isolation rate for traced cases (blue), isolation rate for non-traced cases (purple), contact-identification ratio (orange), and growth rate of the daily vaccination (red). For the sensitivity analysis, parameter ranges were generated based on ±25% variability around baseline values.

**Table 6. The final value and range of PRCC of model inputs considering different model outputs. There is no range if the target model output is the peak number of severe patients as there is a one-time point of it.**

| Model input | Model output | | |
|---|---|---|---|
| | Cumulative confirmed cases | Cumulative deaths | Peak severe patients |
| Outbreak recognition timing | 0.90 [0, 0.93] | 0.90 [−0.19, 0.93] | 0.88 |
| Impact of social distancing | −0.93 [−0.93, 0.01] | −0.91 [−0.91, 0] | −0.85 |
| Isolation rate for traced cases | −0.28 [−0.28, 0] | −0.32 [−0.32, 0.04] | −0.27 |
| Isolation rate for non-traced cases | −0.89 [−0.89, −0.26] | −0.88 [−0.88, 0.2] | −0.81 |
| Contact-identification ratio | −0.01 [−0.29, −0.01] | −0.01 [−0.01, 0.01] | −0.02 |
| Growth rate of the daily vaccination | −0.31 [−0.31, 0] | −0.33 [−0.33, 0] | −0.16 |

vaccination and quarantine measures, which our results support by showing that strong NPIs combined with rapid vaccination significantly reduce outbreak severity [13,14].

Unlike prior studies that focused on single vaccination strategies, our study evaluates different vaccine rollout speeds and prioritization methods. While Mohanty et al. and Costantino et al. highlighted the role of vaccination in outbreak control [17,18], our results suggest that prioritizing high-transmission age groups is more effective than prioritizing high-risk older adults, particularly in scenarios with limited NPIs. Additionally, our study expands upon Ohkusa et al. by showing

that mass vaccination is essential when outbreak recognition is delayed, reinforcing the importance of both proactive and reactive strategies [15]. By incorporating Korea-specific demographic data, healthcare capacity constraints, and a logistic vaccine administration model, our analysis provides a more context-specific policy recommendation while contributing to broader smallpox preparedness research.

The model simulation showed a rapid increase in confirmed cases upon initial detection, followed by a gradual increase in the number of severely ill patients. By 2024, the ROK plans to expand the number of negative-pressure isolation beds to 3500 to respond to emerging infectious diseases [34]. According to the baseline scenario results, even in the worst case within the 95% PI, the peak administered to severely ill patients was 1800 (Fig 4), indicating that the current plan should prevent a shortage of beds. However, in an extreme worst-case scenario, where the vaccination rate is low and the impact of social distancing is weak (upper row in Table 2), the number of administered severe cases could reach up to 7600, posing a significant risk. However, if the effectiveness of social distancing reaches at least 0.6, dangerous situations are avoided. Therefore, the results suggest that a minimum level of NPIs required during a smallpox outbreak. This level was measured to be near the level of social distancing stage 2 in previous COVID-19 studies and is not an unrealistic measure in a bioterrorism situation where nationwide interventions would be stricter.

The parameter sensitivity analysis results provided additional support for the basic model simulation outcomes. The analysis revealed that both deaths and confirmed cases were highly sensitive to recognition timing, impact of social distancing, and isolation rate for non-traced cases (Fig 10 and Table 6). This underscores the significant role of NPIs in achieving herd immunity through vaccination.

Fig 7 represents the estimated number of individuals requiring immediate vaccination following outbreak recognition, serving as the primary basis for determining the minimum vaccination targets. Given that new exposures continue to occur after outbreak recognition, the additional required vaccinations are illustrated in Fig 8, which captures the cumulative vaccination scope over time. Although these targeted vaccinations constituted only ~0.1% of the total vaccination campaign, they were concentrated in the first week to ensure that the most at-risk individuals received timely immunization. This approach aligns with real-world outbreak control strategies, where early vaccination of high-risk contacts is essential to prevent further spread. The results of this study provide a quantitative framework for determining vaccination distribution points and operational strategies during the early containment phase.

Determining vaccine priorities is challenging because of various social, economic, and ethical issues. Similar problems have been encountered during the COVID-19 pandemic. Prioritizing vaccination for the elderly, who have higher severity/fatality rates, was the best strategy for minimizing deaths when social distancing measures were in place to reduce the effective reproduction number to approximately one [35,36]. However, if the effective reproduction number increases—for example, due to a higher contact rate, an increased probability of successful disease transmission, or a lower isolation rate of infectious individuals—prioritizing the elderly would be less effective than vaccinating younger adults in minimizing deaths. The study was based on the original strain of COVID-19, which had a basic reproductive number of approximately three, roughly half of that of smallpox. This implies that strong NPIs can effectively suppress the spread. Conversely, for smallpox, for which moderate levels of NPIs were not sufficient to control the spread, prioritizing vaccination for groups with higher transmission rates was more effective in reducing deaths. Prioritizing the elderly yielded the least favorable results. If the vaccination history of the elderly, which was not considered in this study, were accounted for, they would likely have a relatively lower severity/fatality rate compared to other age groups, further diminishing the effectiveness of the first vaccination strategy. Several studies have emphasized the importance of prioritizing populations based on age, occupation, and health status to achieve the greatest public health impact. For instance, the UK's Joint Committee on Vaccination and Immunisation recommended prioritizing older adults, frontline health and social care workers, and individuals with underlying health conditions during the initial vaccine rollout [37]. This approach aimed to protect those at the highest risk of severe disease while maintaining healthcare system functionality. Similarly, Canada's National Advisory Committee on Immunization advised prioritizing residents and staff of long-term care facilities, adults aged 70 and older,

and frontline healthcare workers to reduce COVID-19-related morbidity and mortality by focusing on the most vulnerable populations [38].

These prioritization frameworks underscore the critical role of targeted vaccination strategies in controlling the spread of infectious diseases and minimizing adverse outcomes. They also highlight the importance of adapting vaccination plans based on the epidemiological and demographic context of a given outbreak. While COVID-19 vaccine prioritization largely focused on protecting high-risk individuals, smallpox, with its higher basic reproductive number and different transmission characteristics, may require a different prioritization strategy. Our findings suggest that in the case of smallpox, prioritizing groups with higher transmission rates rather than solely focusing on vulnerable populations could be more effective in reducing overall mortality.

The limitations of this study are as follows. For social contacts, we only used estimates based on close contacts. Age-specific severity rates were derived from data that included both vaccinated and unvaccinated individuals, which may differ from the actual values. We did consider the smallpox vaccinations administered in the ROK until the early 1970s and rather assumed that all population groups were susceptible. However, despite the lack of vaccine effectiveness against infection, the elderly might have a lower severe/fatality rate than other age groups due to their vaccination history. Finally, although the smallpox vaccine can have significant side effects, we did not incorporate these side effects into our model. This was because vaccination coverage was fixed. Future studies will focus on analyzing optimal vaccination strategies, taking into account side effects, spatial heterogeneity, and regional lockdowns.

## Conclusion

Based on the findings of this study and considering realistic intervention scenarios and outbreak situations, we propose an appropriate number of isolation facilities for severely ill patients and the necessary level of initial social distancing. Various simulations have highlighted the critical importance of early detection and rapid responses to mitigate the impact of small-pox outbreaks. These results underscore the need for robust preparedness plans that include vaccination and NPIs.

Our study emphasizes the importance of strategic vaccination prioritization and the role of NPIs in controlling outbreaks. The insights gained from this study provide valuable guidance to public health officials and policymakers in preparing for and responding to potential biosecurity threats and emerging infectious diseases. The critical importance of early detection, rapid response, and comprehensive preparedness cannot be overstated when safeguarding public health.

The overall framework of this study applies to smallpox and other emerging infectious diseases that may spread to humans in the future. By incorporating parameters similar to those applied in this study, response strategy scenarios can be developed for diseases that can be controlled using currently available vaccines. Conversely, for novel diseases with significant time requirements for vaccine development, this framework can be adapted to simulate the post-declaration phase responses.

## Author contributions

**Conceptualization:** Youngsuk Ko.

**Data curation:** Youngsuk Ko, Jin Ju Park, Eun Jung Kim.

**Formal analysis:** Youngsuk Ko, Yubin Seo, Jin Ju Park, Eun Jung Kim, Jong Youn Moon, Tark Kim, Joong Sik Eom, Hong Sang Oh, Arim Kim, Jin Yong Kim, Jacob Lee, Eunok Jung.

**Funding acquisition:** Jacob Lee, Eunok Jung.

**Investigation:** Youngsuk Ko, Yubin Seo, Jin Ju Park, Eun Jung Kim, Tark Kim, Joong Sik Eom, Hong Sang Oh, Arim Kim, Jin Yong Kim, Jacob Lee.

**Methodology:** Youngsuk Ko.

**Project administration:** Jacob Lee, Eunok Jung.

**Resources:** Youngsuk Ko.

**Software:** Youngsuk Ko.

**Supervision:** Jacob Lee, Eunok Jung.

**Validation:** Youngsuk Ko.

**Visualization:** Youngsuk Ko.

**Writing – original draft:** Youngsuk Ko.

**Writing – review & editing:** Youngsuk Ko, Yubin Seo, Eunok Jung.

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
