## [Decision Letter · Decision Letter 0]

15 Jan 2025

Dear Dr. Jung,

Thank you for submitting your manuscript to PLOS ONE. After careful consideration, we feel that it has merit but does not fully meet PLOS ONE’s publication criteria as it currently stands. Therefore, we invite you to submit a revised version of the manuscript that addresses the points raised during the review process.

We look forward to receiving your revised manuscript.

Kind regards,

Amani Abu-Shaheen, MPH

Academic Editor

PLOS ONE

Journal Requirements:

“This research was supported by the Government-wide R&D Fund Project for Infectious Disease Research (GFID), Republic of Korea (grant No. HG23C1629). This work was supported by the Research Program funded by the Korea Disease Control and Prevention Agency (정책, 150).”

“This research was supported by the Government-wide R&D Fund Project for Infectious Disease Research (GFID), Republic of Korea (grant No. HG23C1629). This work was supported by the Research Program funded by the Korea Disease Control and Prevention Agency (정책, 150).”

“This research was supported by the Government-wide R&D Fund Project for Infectious Disease Research (GFID), Republic of Korea (grant No. HG23C1629). This work was supported by the Research Program funded by the Korea Disease Control and Prevention Agency (정책, 150).”

Comments from PLOS Editorial Office: We note that one or more reviewers has recommended that you cite specific previously published works. As always, we recommend that you please review and evaluate the requested works to determine whether they are relevant and should be cited. It is not a requirement to cite these works. We appreciate your attention to this request.

Reviewers' comments:

Reviewer's Responses to Questions

**Comments to the Author**

1. Is the manuscript technically sound, and do the data support the conclusions?

Reviewer #1: Yes

Reviewer #2: Yes

Reviewer #3: Partly

Reviewer #4: Yes

2. Has the statistical analysis been performed appropriately and rigorously?

Reviewer #1: Yes

Reviewer #2: Yes

Reviewer #3: Yes

Reviewer #4: No

3. Have the authors made all data underlying the findings in their manuscript fully available?

Reviewer #1: Yes

Reviewer #2: No

Reviewer #3: Yes

Reviewer #4: No

4. Is the manuscript presented in an intelligible fashion and written in standard English?

Reviewer #1: Yes

Reviewer #2: Yes

Reviewer #3: No

Reviewer #4: Yes

Reviewer #1: This study constructs a mathematical model for epidemic simulation, which provides a new reference for the prediction and response of smallpox epidemics under the background of bioterrorism. The overall logic is reasonable, the argument is clear, and the conclusion is relatively reliable. It is recommended to add a brief introduction to the existing relevant mathematical models and the innovation of this model in the introduction section.

Reviewer #2: I have reviewed this manuscript from the perspective of a mathematical epidemiologist who has experience in infectious disease modelling, though with no prior experience in modelling the transmission dynamics of smallpox.

Please see the attached file for comments.

Reviewer #3: Dear Authors,

This manuscript is a simulated study based on the SIR model. It uses a mathematical model to analyze the smallpox epidemic, including the control measures. The topic is not interesting for public health since this disease has not been an outbreak since 1980. The introduction should be more convincing as to why the authors chose smallpox in this study. The evidence of small outbreaks may help to improve the introduction. The authors should be concerned about using some specific words without definitions, such as cases, patients, isolated patients, ring or mass vaccinations, and confirmed cases. These words have their definitions. The intervention strategies are not clear on how to apply them.

I have more comments, detailed below.

1. What are the meanings of ring or mass vaccinations in this study? The authors used a mathematical model (ODE fashion) via the Tau-leaping method. Normally, ring vaccination is applied for spatial models or for those who have close contact with infected cases via an individual-based model. I would suggest using another word.

2. “Using this value and the next-generation matrix method, the basic reproductive number through close contact was calculated to be approximately four (16).” It is not clear how to define close contact or social contact.

3. Please separate the control measures into a section so that they can be understood the results. The vaccination strategies should be clear in this section.

4. For “Baseline model simulation scenario” for phase 2, what happens if the duration is more than three days? I think three days is very short to prepare for the next phase.

5. “The number of exposed hosts was set to 100 and proportionally distributed according to the population ratio of each group.” How does this set the initial condition, is it constant over 1000 simulations or randomly based on some function, please clarify.

6. Provide more detail on “The age groups with higher transmission risk were calculated based on the contact rate and the probability of successful disease transmission, and the order was as follows: age groups 4 (15-19 year), 3 (10-14 year), 9 (40-44 year), 8 (35-39 year), 7 (30-34 year), 6 (25-29 year), 10 (45-49 year), 11 (50-54 year), 5 (20-24 year), 2 (5-9 year), 12 (55-59 year), 13 (60-64 year), 1 (0-4 year), 14 (65-69 year), 15 (70-74 year), 16 (75+ year).”, the authors may show in the supplement.

7. Results: Baseline scenario simulation, please provide the symbol of compartments to make it easy to track the results. I would be more understanding of the baseline simulation if the authors could provide the results of no control.

8. Figure 2, please highlight the phases 1-3.

9. “In addition to the baseline scenario, the results for different initial numbers and logistic growth rates for vaccinations and the varying effects of social distancing on contact reduction are listed in Tables 3–5.” What do the authors mean by different initial numbers? To be clear, the authors should provide the figure for vaccinations.

10. Rewrite “Panels A and B show the confirmed cases, and panels C and D show the number of deaths.” To correspond with the Figure 3.

11. Tables 3-5 can be combined and presented in one bar plot. Those tables can move to supplementary.

12. There is a lag on how to calculate probability. Revise the figure caption to be clearer, i.e., what is D+28.

13. Figure 5: it is not clear how to define “the mean daily vaccination number”.

14. Figure 6: Is it cases or patients?

15. Vaccine prioritization should be clear in the method.

16. For PRCC, the authors should provide more detail of each parameter’s link with the mathematical model.

17. Lines 360-367 could not be understood without the clear methods.

18. What the authors mean by this “However, if the effective reproduction number increases”, which parameters increase?

19. Lines 374-381, there is a lot of work doing vaccine prioritization using COVID-19 as a case study. The authors can discuss more on this issue.

Reviewer #4: The work will be helpful for the future researchers who are working in the same direction but it has main limitations on the modeling aspects. The authors should focus on the modeling. How the research will be helpful for the community who are working on the mathematical modeling, statistical approach etc.

The COVID-19 pandemic has already spread throughout the world and the people are aware about the diseases and they are using precautions about the pandemic. But, still the covid-19 is spreading very quickly. There are major comments before considering the second round revision.

---- The abstract is a little thin and does not quite convey the vibrancy of the findings and the depth of the main conclusions. The authors should please extend this somewhat for a better first impression.

---- The manuscript lacks motivation. Author needs to include the motivation of the study.

-----To stop the spread of the diseases vaccine is needed. But, in absence of the vaccine people must have maintain the social distancing. In order to maintain the social distancing must obey the modeling rule. The introduction need to be improved by incorporating some recent references of COVID-19 pandemic. To do so, I suggest some modeling work must be included in the references: "Modeling and forecasting the COVID-19 pandemic in India, Chaos, Soliton & Fract. 139 (2020) 110049", “Mathematical modeling of the COVID-19 outbreak with intervention strategies, Results in Physics, 2021, 104285”, "Forecasting the daily and cumulative number of cases for the COVID-19 pandemic in India, Chaos, 30(7) (2020) 071101", "A mathematical model for COVID-19 transmission dynamics with a case study of India, Chaos, Soliton & Fract. 140 (2020) 110173".

---- In this context an important factor must be include in this study, that is, the impact of the effect of media. How the COVID-19 dynamics has been changed due to incorporation of the media related awareness like use of face masks, non-pharmaceutical interventions, hand sanitization, etc. The authors must include the manuscript, “Impact of social media advertisements on the transmission dynamics of COVID-19 pandemic in India, Journal of Applied Mathematics and Computing (2021)” "Dynamics of the COVID-19 pandemic in India, (2021) arXiv:2005.06286v2." to study the effect of media.

----Is there any experimental data to validate the mathematical model ? The authors at least describe the basic reproduction number R_0 and its impact on covid-19 pandemic. The basic reproduction number R_0 is one of the most crucial quantities in infectious diseases, as R_0 measures how contagious a disease is. For R_0 < 1, the disease is expected to stop spreading, but for R_0 = 1 an infected individual can infect on an average 1 person, that is, the spread of the disease is stable. The disease can spread and become epidemic if R_0 must be greater than 1. In this context the authors include the reference "Mathematical analysis of the global dynamics of a HTLV-I infection model, considering the role of cytotoxic T-lymphocytes, Math. Comput. Simul. 180(2021) 354-378."

----Some references contain errors and inconsistent formatting. It is difficult to give credit to research if even elementary aspects of the work are not error free. This should be corrected with care and love to detail.

----The manuscript is comprehensive, and I have enjoyed learning about the presented results. I find that the manuscript is written with very poor english and the presentation is not good, and I am in principal in favor of publication, although the following comments should nevertheless be accommodated in one major revision.

**Do you want your identity to be public for this peer review?** For information about this choice, including consent withdrawal, please see our Privacy Policy

Reviewer #1: No

Reviewer #2: No

Reviewer #3: No

Reviewer #4: No

---

## [Author Response · Author response to Decision Letter 1]

28 Apr 2025

Thank you for valuable comments. We have attached response file as word file.

---

## [Decision Letter · Decision Letter 1]

15 Aug 2025

Dear Dr. Eunok Jung,

Thank you for submitting your manuscript to PLOS ONE. After careful consideration, we feel that it has merit but does not fully meet PLOS ONE’s publication criteria as it currently stands. Therefore, we invite you to submit a revised version of the manuscript that addresses the points raised during the review process.

Based on the reviewers’ feedback, I find that the manuscript is scientifically sound and presents valuable findings. However, Reviewer 2 raised some points that, while relatively minor, should be addressed to further strengthen the clarity and completeness of your work.

Please carefully review the comments from Reviewer 2 and provide a point-by-point response, making the necessary revisions in the manuscript.

We look forward to receiving your revised manuscript.

Kind regards,

Sara Hemati

Academic Editor

PLOS ONE

Journal Requirements:

Reviewers' comments:

Reviewer's Responses to Questions

**Comments to the Author**

Reviewer #2: (No Response)

Reviewer #3: All comments have been addressed

2. Is the manuscript technically sound, and do the data support the conclusions?

Reviewer #2: Yes

Reviewer #3: Yes

3. Has the statistical analysis been performed appropriately and rigorously?

Reviewer #2: Yes

Reviewer #3: N/A

4. Have the authors made all data underlying the findings in their manuscript fully available?

Reviewer #2: Yes

Reviewer #3: Yes

5. Is the manuscript presented in an intelligible fashion and written in standard English?

Reviewer #2: Yes

Reviewer #3: Yes

Reviewer #2: This submission is a revision of a study that presents a mathematical model to analyse potential smallpox epidemics, considering factors like age, contact patterns and intervention strategies (contact tracing, targeted vaccination and mass vaccination).

I have uploaded my review as an attachment. Please see that report.

Reviewer #3: The authors have addressed the comments well, except comment 3.9. However, the method is clearer and can be understood in phases 1-3.

**Do you want your identity to be public for this peer review?** For information about this choice, including consent withdrawal, please see our Privacy Policy

Reviewer #2: No

Reviewer #3: No

---

## [Author Response · Author response to Decision Letter 2]

8 Sep 2025

Comment 1.

I did not see any documentation provided with the repository to outline what the purpose of each file was. Addition of a README type file that provides that information I think would be helpful. That can also state the version of MATLAB used for the study and any prerequisite packages needed to run the study code.

Response 1.

We thank the reviewer for this helpful suggestion. As recommended, we have added a README.txt file to the repository. The README.txt provides a description of the purpose of each code file, specifies the MATLAB version used in the study, and lists the example data included. We believe this addition will make it easier for readers to reproduce and understand the code.

README.txt

This repository contains the MATLAB code used in the study.

Note: The code was run using MATLAB R2024b.

[Main scripts and functions]

func_sto_v1.m

Function file implementing the tau-leaping method.

scp_sto_main.m

Main script that generates stochastic simulation runs.

scp_plot_check.m

Visualization script for the main outputs.

Requires pre-generated data (e.g., multishot_2_1_2.mat containing baseline simulation runs).

scpaux_tauleap_v1.m

Auxiliary script for setting up matrices and vectors used in the main code.

[Example data]

multishot_2_1_2.mat

Example dataset containing baseline simulation runs, used for plotting and checking outputs.

Comment 2.

There is a lack of comments in each code script. That makes it challenging to understand the purpose of blocks of code. A suggestion to add comments to the code scripts to help guide the user.

Response 2.

To address the reviewer’s concern, we have added comments to the main script (scp_sto_main.m) to clarify the structure and purpose of key code blocks. In addition, we have included a README file that describes the purpose of each script and how they are used, along with information on the MATLAB version and example data provided. We believe these additions will guide readers sufficiently in running and understanding the code, while keeping the repository streamlined.

Comment 3.

File ‘scp_plot_check.m’: Ran without error for me as long as I had first run the file ‘scp_sto_main.m’. However, if running the file with a clear workspace, the process errored as the prerequisite variables had not been generated. Please can amendments be made to avoid that situation (e.g. Another MAT file can be loaded that has example data to produce the plot set)?

Response 3.

As suggested, we have included example data (baseline scenario) as a .mat file (multishot_2_1_2.mat). This allows the reviewer to load the data directly and test the plotting script without first running the main simulation.

Comment 4.

File ‘scpaux_tauleap_v1.m’: Also got errors attempting to run this file, even after running the main script (specifically, programme terminated on line 31 as the variable ‘scpaux_tauleap_v1’ had not been defined). Please check and provide extra information on how this file should be used by the user.

Response 4.

We tested the file using the repository as currently provided, and it runs without errors in our environment. However, we cannot guarantee compatibility with MATLAB versions prior to R2024b. We kindly ask the reviewer to check the MATLAB version being used and re-run the test accordingly.

Comment 5.

For reproducibility purposes and to permit verification of model applicability, I believe more information needs to be provided on the parameterisation of social contacts, close contacts and transmission risk in the model.

I consider it reasonable that the main text reports the parameter values used. i.e. Overall basic reproduction number of six. Probability of successful disease transmission through close contact set to 60% and through social contact set to 10%. Social contacts occur at four times the frequency of close contacts.

However, I think what is then also needed is an expanded mathematical description as a section in the Supporting Information to describe the calibration of the close contact and social contact contributions to the overall reproduction number. i.e. Given “the basic reproductive number through close contact was calculated to be approximately four”, how the social contact parameters set at the stated relative contact frequency (compared to close contacts) and the successful disease transmission through social contact values results in the contribution of social contacts to the overall basic reproduction number being two.

Response 5.

We thank the reviewer for this helpful comment. In our study, the social contact parameters were chosen as simplifying assumptions to calibrate the overall reproduction number to approximately six, consistent with historical estimates for smallpox. Specifically, we assumed that social contacts occur at about four times the frequency of close contacts (excluding household contacts) and set the transmission probability through social contact to 10%. Together with the 60% probability for close contacts, this results in an estimated R₀ of ≈4 from close contacts and ≈2 from social contacts, yielding a total R₀ of ≈6.

To clarify this rationale, we have revised the Methods section. At the end of the relevant paragraph, we now explicitly state (L181):

“These parameter values were chosen as simplifying assumptions to reproduce an overall basic reproductive number of approximately six, rather than as precise empirical estimates of contact-specific transmission probabilities.”

Comment 6.

Helpful information has now been added into the revised manuscript to describe the sensitivity analysis. That information has all been added into the Results section. In my opinion, a substantial part of that added text is methods (L410-437). For ease of reader reference, the text describing the methodology for conducting the sensitivity analysis I would consider to be better placed at the end of the Methods section. The results subsection on the sensitivity analysis can then instead open with a high-level summary/reminder to the reader of what the purpose of the sensitivity analysis is in the context of this study.

Response 6.

In the revised manuscript, we have restructured the sensitivity analysis section to improve clarity. Specifically, the methodological details (L410–437) describing the use of Latin hypercube sampling, PRCC, and the parameter ranges have been moved from the Results section to the end of the Methods section under the subheading “Sensitivity Analysis”. The Results subsection on sensitivity analysis now begins with a high-level summary that emphasizes the purpose of the analysis and then presents the findings, with a reference directing the reader to the Methods section for full details.

Comment 7.

I provided the following comment in my original review: “Recommend these are expanded to provide sufficient details so the features of each figure can be understood by the reader without having to refer to the main text. To give a couple of examples, I would suggest removing from the main text and instead placing as part of the relevant figure caption: L222-226 describing Figure 2, instead have as part of the Figure 2 caption; L322-324 describing Figure 7, instead have as part of the Figure 7 caption. Similar changes can be made to the remaining figure captions.”

The authors have actioned the couple of examples I suggested. However, I have not seen similar edits made to the other figure captions. I still view that for the other figure captions it would be helpful for more information to be given (for purpose of enabling the features of each figure to be understood by the reader without having to refer to the main text).

Response 7.

Following the suggestion, we have revised the figure captions to make them more self-contained, so that the main features of each figure can be understood without needing to refer to the main text:

Figure 5. Outbreak outcomes from the baseline scenario simulations. Panels (A) and (B) show the distribution of confirmed cases by age group and in total, respectively. Panels (C) and (D) present the corresponding distributions of deaths. Results are displayed as violin plots, which illustrate both the variability and the probability density of outcomes across stochastic simulation runs.

Figure 6. Outbreak outcomes under different assumptions for vaccination growth rate and contact reduction, given a baseline initial vaccination number of 1,000. Results are shown on a log scale for confirmed cases (A), deaths (B), and peak severe patients (C). Each color denotes a different vaccination growth rate, while the horizontal axis indicates the level of contact reduction.

Figure 8. Vaccination numbers in the baseline scenario simulations. Panel (A) shows the mean daily number of vaccinations for ring vaccination (magenta) and mass vaccination (black), with the inset highlighting the early outbreak period. Panels (B) and (C) present the probability distributions for the total number of doses administered and the duration of the ring vaccination campaign, respectively, across stochastic simulation runs.

Figure 9. Simulation results under different vaccine prioritization strategies. Panel (A) shows daily confirmed cases, and Panel (B) shows the number of severe patients over time. The scenarios compared are: baseline (uniform vaccination, dotted blue), ascending order of age (solid orange), descending order of age (solid yellow), transmissibility-based prioritization (dashed purple), and fatality-based prioritization (dashed green).

Figure 10. Absolute partial rank correlation coefficients (PRCC) over time for key model parameters. Panel (A) shows PRCC values with respect to cumulative confirmed cases, and Panel (B) shows PRCC values with respect to cumulative deaths. Each color denotes a different input parameter: outbreak recognition timing (yellow), impact of social distancing (green), isolation rate for traced cases (blue), isolation rate for non-traced cases (purple), contact-identification ratio (orange), and growth rate of the daily vaccination (red). For the sensitivity analysis, parameter ranges were generated based on ±25% variability around baseline values.

Comment 8.

This new figure helpfully summarises variability in outcomes under different assumptions. To enable the reader to see the underlying outcome distributions, a request to please change the box plots to be raincloud plots.

Response 8.

We thank the reviewer for the suggestion. In this figure, nine separate distributions are presented, and the main quantitative findings are already provided in the table and described in the text. Changing to raincloud plots would make the visualization overly complex giving disproportionate emphasis to this secondary result, which is not among the central findings of the study. In addition, since the y-axis is shown on a logarithmic scale, density shapes in raincloud plots would be visually distorted and difficult to interpret. For these reasons, we believe it is more appropriate to retain the current box plot format, which conveys the variability in a clear and concise manner.

Comment 9.

The authors have made edits to help distinguish the different line profiles from one another. These have helped, though I think further changes would be useful to make clear to the reader the differences in outcomes between the different scenarios.

The age-based order scenarios are both solid lines. The not age-based order scenarios are both dashed lines. To help the reader distinguish between the lines in these respective pairs, a suggestion to add different marker styles to the lines (e.g. crosses, filled circles, unfilled circles, triangles)

Could present the daily case numbers and severe patient numbers on a log scale

Adding two additional panels to the figure that show the difference between each scenario compared to the baseline strategy.

• Panel C showing the differences compared to the baseline strategy for ‘Daily confirmed cases’

• Panel D showing the differences compared to the baseline strategy for ‘Severe patients’

Response 9.

We thank the reviewer for these detailed suggestions. At the current level of complexity, we believe the line profiles are distinguishable without additional markers, and adding markers would make the figure visually cluttered given the number of curves displayed. Presenting the outcomes on a logarithmic scale would also narrow the visual differences between scenarios, reducing the clarity of the comparison. Furthermore, the relative differences between scenarios are already summarized quantitatively in Table 5 using odds ratios. To avoid redundancy and maintain figure readability, we have chosen to retain the current figure format without additional panels.

Comment 10.

Figure 10 (formerly Figure 7)

The split of this figure into two separate panels I found has made the overall figure less busy. I do have other concerns regarding accessibility, due to the line styles all being the same and some of the colours being similar. I recommend edits are made to help the reader distinguish between the different line profiles. One suggestion is to use a monochromatic colour scheme – a single colour for all the line profiles with each line profile having a different shading intensity (so that across the scenarios the line profiles range from a light shade to a dark shade).

Response 10.

We thank the reviewer for the suggestion. In this figure, each line represents a fundamentally different model input parameter (e.g., outbreak recognition timing, contact identification ratio, social distancing, vaccination growth rate, isolation rates). For this reason, we believe it is clearer to maintain distinct colors for each line, so that the reader can directly associate each profile with a specific intervention or epidemiological factor. Using a monochromatic color scheme with shading intensity would risk obscuring these distinctions, especially given the number of parameters included. To address accessibility, we have clarified the color-to-parameter mapping explicitly in the caption, ensuring that readers can readily distinguish the line profiles.

Comment 11.

L29-30: “This study develops a mathematical model to evaluate outbreak scenarios and the effectiveness of reactive intervention strategies in controlling transmission.” Please add that this evaluation is applied to the Republic of Korea.

Response 11.

We appreciate the reviewer’s suggestion. As recommended, we have revised the sentence in the Abstract to clarify the study setting:

“This study develops a mathematical model to evaluate outbreak scenarios and the effectiveness of reactive intervention strategies in controlling transmission, with application to the Republic of Korea.”

Comment 12.

L95: “negative-pressure isolation beds to 3,500 to response to Disease-X”. Amend “response” to “respond”.

Response 12.

Corrected.

Comment 13.

L244: Please add the version number of MATLAB that the study code was run with.

Response 13.

Corrected (2024b).

Comment 14.

The suggested subheadings listed in the Response to Reviewers I view as sensible. However, the subheadings in the revised text appear to slightly differ. I suggest amending them to match the list below stated by the authors in the Response to Reviewers document:

• Baseline Scenario Simulation: Outbreak Outcomes

• Baseline Scenario Simulation: Impact of Outbreak Recognition on Outbreak Size

• Baseline Scenario Simulation: Vaccine Adminis

---

## [Decision Letter · Decision Letter 2]

13 Oct 2025

Dear Dr. Eunok Jung,

Thank you for submitting your manuscript to PLOS ONE. After careful consideration, we feel that it has merit but does not fully meet PLOS ONE’s publication criteria as it currently stands. Therefore, we invite you to submit a revised version of the manuscript that addresses the points raised during the review process.

https://journals.plos.org/plosone/s/submission-guidelines#loc-laboratory-protocols . Additionally, PLOS ONE offers an option for publishing peer-reviewed Lab Protocol articles, which describe protocols hosted on protocols.io. Read more information on sharing protocols at https://plos.org/protocols?utm_medium=editorial-email&utm_source=authorletters&utm_campaign=protocols .

We look forward to receiving your revised manuscript.

Kind regards,

Sara Hemati

Academic Editor

PLOS ONE

Journal Requirements:

**Comments from PLOS One Editorial Office:**

Thank you very much for providing your previous responses to the scientific concerns raised in this manuscript, please address Reviewer 2s comments on the revised manuscript.

We note that this manuscript focuses on the risks posed from a theoretical smallpox outbreak, potentially from a bioterrorism incident. Overstating the risk to the public posed from possible bioterrorism incidents could lead to sensationalism and alarm.

In the case of your manuscript we note that the model presented in this paper does not appear to be designed to simulate an outbreak caused by bioterrorism specifically, but rather discusses general outbreaks, and the authors also discuss how their model could be applied to other outbreak scenarios. Given this, we have concerns that the substantial focus on bioterrorism throughout the manuscript is not justified.

We therefore recommend the following revisions to the manuscript prior to acceptance:

- Title: Smallpox *outbreak* scenarios and reactive intervention protocols: A mathematical model-based analysis applied to the Republic of Korea

- Abstract line 1: Smallpox, caused by the variola virus, remains a potential *biosecurity* threat despite its eradication.

- Line 72: However, the presence of these viral stocks poses *biosecurity concerns, including* the potential risk of bioterrorism. Given the lack of widespread immunity in the current global population, the *accidental or* deliberate release of the smallpox virus could lead to a catastrophic outbreak.

- Line 127: Given the *potential biosecurity* threat *posed by* smallpox

- Line 137: emphasizing the need for comprehensive preparedness in the face of potential *outbreaks*.

- Line 567: The insights gained from this study provide valuable guidance to public health officials and policymakers in preparing for and responding to potential *biosecurity* threats and emerging infectious diseases

Reviewers' comments:

Reviewer's Responses to Questions

**Comments to the Author**

Reviewer #2: (No Response)

2. Is the manuscript technically sound, and do the data support the conclusions?

Reviewer #2: Yes

3. Has the statistical analysis been performed appropriately and rigorously?

Reviewer #2: Yes

4. Have the authors made all data underlying the findings in their manuscript fully available?

Reviewer #2: Yes

5. Is the manuscript presented in an intelligible fashion and written in standard English?

Reviewer #2: Yes

Reviewer #2: I thank the authors for the additional revisions they have made to the manuscript. Most of my previous comments I feel have been addressed.

The following items in my view still require attention.

Figure 10: I unfortunately do not consider the authors rebuttal to my previous comment raised on Figure 10 as sufficient – the accessibility issues on this figure I believe remain (to a lesser extent also in Figure 9).

Specifically, on the response “To address accessibility, we have clarified the color-to-parameter mapping explicitly in the caption, ensuring that readers can readily distinguish the line profiles”, if a reader has a colour-blindness condition that means they are not able to distinguish between the line colours, then the line colour information in the caption does not help in that situation (as the lines all look similar).

I therefore restate my recommendation that edits are made to this figure to help the reader distinguish between the different line profiles (given the number of lines meaning each line having a unique line style may also not be possible, that is why a monochromatic colour scheme may be a potential solution).

Comments within code scripts: Within the code files associated with the study, helpful comments have been added to the main script and a guiding README file added. Nonetheless, I disagree with the authors assessment that “We believe these additions will guide readers sufficiently in running and understanding the code” as the supporting scripts and function files lack comments. There are consequently multiple files where additional support can still be provided to the reader. Therefore, my previous comment stating “There is a lack of comments in each code script” is still to be fully resolved.

**Do you want your identity to be public for this peer review?** For information about this choice, including consent withdrawal, please see our Privacy Policy

Reviewer #2: No

---

## [Author Response · Author response to Decision Letter 3]

26 Nov 2025

Editor’s comments

We therefore recommend the following revisions to the manuscript prior to acceptance:

- Title: Smallpox *outbreak* scenarios and reactive intervention protocols: A mathematical model-based analysis applied to the Republic of Korea

- Abstract line 1: Smallpox, caused by the variola virus, remains a potential *biosecurity* threat despite its eradication.

- Line 72: However, the presence of these viral stocks poses *biosecurity concerns, including* the potential risk of bioterrorism. Given the lack of widespread immunity in the current global population, the *accidental or* deliberate release of the smallpox virus could lead to a catastrophic outbreak.

- Line 127: Given the *potential biosecurity* threat *posed by* smallpox

- Line 137: emphasizing the need for comprehensive preparedness in the face of potential *outbreaks*.

- Line 567: The insights gained from this study provide valuable guidance to public health officials and policymakers in preparing for and responding to potential *biosecurity* threats and emerging infectious diseases

Response to editor.

All suggested changes have been fully incorporated into the revised manuscript.

Reviewer #2:

Comment 2-1.

Figure 10: I unfortunately do not consider the authors rebuttal to my previous comment raised on Figure 10 as sufficient – the accessibility issues on this figure I believe remain (to a lesser extent also in Figure 9).

Specifically, on the response “To address accessibility, we have clarified the color-to-parameter mapping explicitly in the caption, ensuring that readers can readily distinguish the line profiles”, if a reader has a colour-blindness condition that means they are not able to distinguish between the line colours, then the line colour information in the caption does not help in that situation (as the lines all look similar).

I therefore restate my recommendation that edits are made to this figure to help the reader distinguish between the different line profiles (given the number of lines meaning each line having a unique line style may also not be possible, that is why a monochromatic colour scheme may be a potential solution).

Response 2-1.

We have revised Figure 10 so that all six line profiles can now be distinguished even in grayscale or color-blind viewing conditions by using a combination of line types (solid, dashed, dotted) and distinct square markers.

Comment 2-2.

Comments within code scripts: Within the code files associated with the study, helpful comments have been added to the main script and a guiding README file added. Nonetheless, I disagree with the authors assessment that “We believe these additions will guide readers sufficiently in running and understanding the code” as the supporting scripts and function files lack comments. There are consequently multiple files where additional support can still be provided to the reader. Therefore, my previous comment stating “There is a lack of comments in each code script” is still to be fully resolved.

I did not see any documentation provided with the repository to outline what the purpose of each file was. Addition of a README type file that provides that information I think would be helpful. That can also state the version of MATLAB used for the study and any prerequisite packages needed to run the study code.

Response 2-2.

As response, we have substantially revised the README.txt. Specifically, we added header comments describing the purpose, inputs, outputs, and dependencies of every function and script. Additionally, we expanded the repository README to include a file-by-file description, execution instructions, required MATLAB version (R2024b), and necessary toolboxes. We believe these revisions provide the level of clarity requested and ensure that readers can now easily understand the structure and purpose of each script in the repository.

========README.txt=============

# MATLAB Code Repository for the Study

This repository contains all MATLAB scripts and function files used in the stochastic simulation analyses presented in the study.

All code was executed using **MATLAB R2024b**.

## Required MATLAB Toolboxes

The following toolboxes are required to run the simulation code:

- **Statistics and Machine Learning Toolbox**

(random number generation, probability distributions)

## How to Run the Code

1. Run **scp_sto_main.m** to generate stochastic simulation outputs.

This script initializes parameters, sets simulation options, and calls the tau-leaping engine.

2. The main script automatically uses:

- **func_sto_v1.m** for the tau-leaping implementation

- **scpaux_tauleap_v1.m** for generating auxiliary matrices required for simulation

3. Once simulation output is generated, run **scp_plot_check.m** to visualize the results.

Example pre-generated data (`multishot_2_1_2.mat`) is included for convenience.

## File-by-File Description

### **1. scp_sto_main.m** — Main script (entry point)

**Purpose:**

Runs the stochastic simulation study using the tau-leaping algorithm.

**Inputs:**

- Parameter settings defined inside the script

- Auxiliary structures loaded/generated by helper functions

**Outputs:**

- `.mat` file containing the full set of stochastic simulation trajectories

- Summary variables used for visualization

**Dependencies:**

- Calls `func_sto_v1.m`

- Calls `scpaux_tauleap_v1.m`

### **2. func_sto_v1.m** — Tau-leaping function

**Purpose:**

Implements the tau-leaping method to approximate the stochastic reaction process at each time step.

**Inputs:**

- Parameter structure

- Current state vector

- Time step size (`tau`)

- Stoichiometric matrices and rate indices from `scpaux_tauleap_v1.m`

**Outputs:**

- Updated system state after one tau step

- Reaction counts for each event type

**Dependencies:**

- Requires outputs generated by `scpaux_tauleap_v1.m`

### **3. scpaux_tauleap_v1.m** — Auxiliary setup script

**Purpose:**

Pre-computes matrices, stoichiometric structures, rate-function indices, and bookkeeping vectors needed for tau-leaping.

**Outputs:**

- A structured set of objects used by both `func_sto_v1.m` and `scp_sto_main.m`

**Notes:**

This script is automatically called by `scp_sto_main.m` and does not need to be run independently.

### **4. scp_plot_check.m** — Plotting and visualization

**Purpose:**

Generates plots and summary diagnostics from completed stochastic simulations.

Used for visual inspection and reproduction of key figures.

**Inputs:**

- `.mat` output generated by `scp_sto_main.m`, or

- Example dataset: `multishot_2_1_2.mat`

**Outputs:**

- Figures and visualization panels for analysis

## Example Data

### **multishot_2_1_2.mat**

Contains a pre-generated set of baseline stochastic simulation outputs.

Useful for testing the plotting script or reproducing examples without running full simulations.

---

## [Editor Report · Decision Letter 3]

1 Dec 2025

Smallpox outbreak scenarios and reactive intervention protocols: A mathematical model-based analysis applied to the Republic of Korea

PONE-D-24-53297R3

Dear Dr. Eunok Jung,

We’re pleased to inform you that your manuscript has been judged scientifically suitable for publication and will be formally accepted for publication once it meets all outstanding technical requirements.

Kind regards,

Sara Hemati

Academic Editor

PLOS ONE

Additional Editor Comments (optional):

Accept
---

## [Editor Report · Acceptance letter]

PONE-D-24-53297R3

PLOS One

Dear Dr. Jung,

I'm pleased to inform you that your manuscript has been deemed suitable for publication in PLOS One. Congratulations! Your manuscript is now being handed over to our production team.

Kind regards,

on behalf of

Dr. Sara Hemati

Academic Editor

PLOS One